# Multidimensional performance assessment, government competition and ecological welfare performance

**Shanhong Li, Yanqin Lv⊙\*, Tianzheng Fan, Ziye Zhang, Gao Feng, Chen Jing**

College of Economics and Management, Xinjiang University, Xinjiang, China

\* 1209238262@qq.com

**Data Availability Statement:** All relevant data are within the paper and its Supporting information files.

## Abstract

Improving the efficiency of converting natural resources into social benefits is an important issue for sustainable development in today's world. Based on this background this paper applies the super-efficient SBM model with non-expected output to measure the ecological welfare performance (EWP) of Chinese provinces from 2005–2019, and explores the relationship between government competition and EWP under different assessment systems. The research results show that government competition under economic performance assessment is self-interested and short-sighted, which can negatively affect ecological welfare performance in the current period as well as in the next four years. In contrast, government competition under the environmental assessment system promotes EWP in both the current and long term, balancing short-term and long-term benefits. The results of the spatial GMM found that government competition under economic performance appraisal can deteriorate EWP in local as well as surrounding areas, but government competition under the environmental assessment system can achieve an increase in local ecological welfare performance and the spillover effect is not significant. To alleviate the limitations of a single appraisal system, this paper incorporates both economic and ecological appraisals into the multidimensional appraisal system. When the weights of both are between 1:9 and 3:7, the government competition under multidimensional performance appraisal can promote both current and longer-term EWP, and achieve its own ecological welfare performance without affecting the surrounding areas.

## 1 Introduction

With China's economic development entering a new normal, the Chinese economy has changed from the original high-speed growth stage to the current high-quality development stage, and transitioned from the original black development model with high energy consumption and high emissions to the green development model with intensification and efficiency [1, 2], but the problems of natural resource scarcity and ecological damage brought by the crude economic development mode are still bottlenecks that plague China's high-quality economic development [3]. At the same time, the concept of sustainable development has been promoted in countries around the world, and scholars have found that economic development

**Funding:** This research is supported by National Social Science Fund of China, Project No.:22XJY036; Xinjiang Autonomous Region Social Science Foundation, Project No.: 21BJY046; Xinjiang University Excellent Doctoral Student Research Innovation Project, Project No: XJU2022BS014. The funders had no role in study design, data collection and analysis, decision to publish, or preparation of the manuscript.

**Competing interests:** The authors have declared that no competing interests exist.

sometimes cause damage to the physical and mental health of residents [4, 5]. In other words, the level of development of people's life cannot be measured by GDP, it is necessary to establish a development concept centered on people's well-being, and promoting people's well-being is the fundamental purpose of development [6]. Ecological environment and social welfare are not two independent systems. As people consume ecological resources as capital inputs to drive economic and social progress, man-made capital is greatly enriched and resource economy becomes the main driver of human welfare. However, when this process reaches a certain level, ecological resources become relatively scarce, and natural capital gradually becomes a major constraint on the improvement of human welfare [7]. How to maximize social welfare with minimal natural resource input, that is, how to improve ecological welfare performance has become a hot topic in China and the world's economic development, and the solution to this problem is also an important extension of sustainable development.

Notably, because local governments have discretionary power in areas such as taxation, regulation, and approval, it is possible for governments to intervene in the transfer of land and the allocation of funds, which can have a significant impact on the welfare and ecological environment [8]. Government behavior is influenced by the performance appraisal system. The central government ranks local officials according to the assessment criteria set, and the higher the ranking, the higher the probability of promotion. This promotion tournament mechanism has given rise to fierce competition among local governments around the appraisal system [9]. There is a "principal-agent" relationship between central governments and the local government, in which local government officials act as agents to achieve national strategic development goals [10]. However, due to the existence of asymmetric information between client and agent, the principal is unable to monitor the agent's effort. When the assessment system is not clear, local governments tend to send signals to their principals through GDP growth, which is the most obvious way [11]. Since the reform and opening up of China, government officials at all levels have been in long-term economic competition for political promotion, which has driven rapid economic growth in their jurisdictions, and local government competition has been attributed by many scholars as an important driving force for China's economic take-off [12–14]. Since the 1990s, economic development has been the focus of government work, and government officials at all levels in China have been engaged in economic competition for political promotion, which has driven rapid economic growth in their jurisdictions, so local government competition has been attributed by many scholars as an important driving force for China's economic takeoff. Government competition under the economic appraisal system will form a path dependence on the crude economic development approach, resulting in the waste of resources, destruction of the environment and neglect of residents' welfare. Even under the "cascade" of governments at all levels, the economic growth pressure exacerbates the vulnerability of the entire ecosystem and economic system [15], which is not conducive to the sustainable and high-quality development of China's economy. With the promotion of the green concept and the transformation of the development mode in China, the Chinese government issued documents such as the Decision of the Central Committee of the Communist Party of China on Some Major Issues Concerning Comprehensively Deepening the Reform in 2013, proposing the policy of "not judging heroes by GDP growth rate" and increasing the proportion of environmental Assessment in the assessment system. Under the change in the appraisal system, local government officials will be more careful to consider the harmonious development between economy, environment and society when competing for the promotion [16]. In summary, the discussion of the relationship between government competition and EWP from the perspective of the appraisal system is of great practical importance.

The marginal contributions of this paper lie in the following three aspects. First, the calculation of EWP by treating environmental pollutants as non-desired outputs is more accurate

than previous scholars' practices. Improvements in measurement methods can better reflect the level of sustainable development of provinces. Second, this paper can effectively explore the impact of government competition on EWP under economic performance assessment and environmental performance assessment, and study its short-term effect, long-term effect and spillover effect. Finally, considering the possible limitations of a single appraisal system, this paper further incorporates both economic and ecological appraisals into the multidimensional performance appraisal system, and adjusts the proportion of two kinds of assessment to tests the optimal multi-dimensional assessment system that can mitigate the negative effects of government competition.

The next sections of this paper are organized as follows. Section 2 presents the literature review. Section 3 describes the theoretical mechanism. Section 4 presents the research methodology and data sources. Section 5 analyzes the empirical results. Finally, Section 6 presents the basic conclusions and policy implications.

## 2 Literature review

### 2.1 Measurement of EWP

The concept of sustainable development was first introduced in 1987, and the term has attracted a great deal of academic attention, leading many scholars to study how to maximize economic development under ecological carrying capacity. However, Daly (1974) argued that economic growth is only a subgoal of the social welfare system, and the purpose of overall social development is to enhance the social welfare of the population, so sustainable development should strike a balance between the consumption of natural resources and the enhancement of social welfare [17]. It is in this context that the idea of EWP was born, Knight (2011) defined it as the efficiency with which ecological resource consumption is translated into the level of social welfare [18]. EWP is a departure from the traditional paradigm of economic growth research and provides a new perspective for sustainable development.

How to measure EWP has become a hot topic in academic circles, and main methods are currently used to measure EWP are ratio method and data envelopment analysis (DEA). Daly (1974) first proposed to measure EWP with the ratio of services to throughput, in which services refer to the level of human welfare from the ecosystem, and throughput includes the natural resources obtained from the ecosystem and the waste discharged to the ecosystem. Although the metric idea points the way for later scholars' research, Daly's ratio method does not give a specific measurement index in practice. Ng (2008) used the ratio of Happy Life Years to Ecological Footprint to measure ecological economic efficiency. Using happiness to characterize social welfare has some limitations, while the HDI combines economic benefits (economic development) and non-economic effects (education, life expectancy), and is a more objective indicator to assess social welfare [19]. Therefore, the ratio of HDI to ecological footprint has been more commonly used by many subsequent scholars to measure EWP [20, 21].

With the development of Data Envelopment Analysis (DEA) models and their wide influence in academic circles, many scholars have conducted a large number of studies by this method instead of the ratio method in recent years. DEA measure EWP compared to the ratio method can make the three systems—economic, social and ecological—more closely linked, and Table 1 summarizes the studies of relevant DEA models for measuring EWP.

**Table 1. A summary of DEA models for measuring EWP.**

| References | DEA model | Objective area | Period |
|---|---|---|---|
| Zhu, et al. (2022) [22] | Two-Stage Super-SBM | 102 countries | 2014 |
| Ehrenstein, et al (2020) [23] | SBM | 151 countries | 2018 |
| Song and Mei(2022) [24] | DEA | 30 Chinese provinces | 2019–2020 |
| Bao, et al. (2023) [25] | SE-SBM | 11 Chinese provinces | 2009–2020 |
| Wang and Feng (2020) [26] | Super-DEA | 30 Chinese provinces | 2006–2018 |
| Teng, et al. (2023) [27] | Super-SBM | 30 Chinese provinces | 2004–2019 |
| Alfonso and Pardo (2016) [28] | CCR and BCC | 11 Colombian cities | 2005–2013 |

## 2.2 Influencing factors of EWP

Ecological Welfare Performance (EWP) is an important indicator for measuring urban sustainable development, which is influenced by various factors including economic, social, and governmental aspects. Research has shown that the level of economic development and population growth are fundamental factors affecting EWP performance. Policymakers can improve EWP by effectively controlling population growth, promoting energy conservation, and emission reduction. Furthermore, factors such as the level of financial development, urbanization, industrial structure, and openness also have a positive impact on EWP [29]. However, fiscal revenue decentralization may inhibit the improvement of EWP. In addition to traditional economic and social factors, environmental factors and technological innovation also have an impact on EWP performance. Green transformation is considered as an emerging strategy to enhance EWP by optimizing resource utilization to coordinate ecological, social, and economic growth, ultimately improving EWP. Therefore, the utilization of renewable energy and technological innovation are also important means to improve EWP [30]. Furthermore, government transparency and industrial agglomeration also have an impact on EWP. A transparent government contributes to improving ecological efficiency, while the impact of industrial agglomeration and technological innovation on EWP varies by region [31, 32]. New urbanization construction is one of the important means to improve EWP. Through adjusting city scale, industrial structure, technological innovation, and dependence on foreign trade, urbanization can effectively improve resource allocation efficiency, promote high-quality development, and enhance EWP. However, government fiscal pressure may restrain the improvement of EWP performance level [33].

## 2.3 Government competition and EWP

Research on the relationship between government competition and ecological welfare performance has mainly focused on three aspects:

Government competition and green development. Scholars mainly use eco-efficiency, green total factor productivity and green development efficiency as proxies, and explore the impact of government competition on the green development. Scholars generally believe that government competition under the economic assessment system inhibits green and high-quality development. In order to achieve rapid economic catch-up, governments rely on the crude development model and compete with each other to lower the level of environmental regulations to attract the entry of high-polluting enterprises, which hinders green and high-quality development [34, 35]. Tang (2021) used provincial data to demonstrate that when both economic and environmental assessments are added to the performance appraisal system, the "bottom-up competition" of government competition changes to "top-

down competition", and the changed government competition promotes eco-efficiency [36]. Some scholars have also pointed out that the weight of economic assessment in the governance system of officials has a direct impact on the effectiveness of environmental governance. Overemphasis on economic performance indicators will not prevent the emergence of "free-rider" behavior among regions, leading to an optimal strategy of environmental non-governance among regions [37].

Government competition and social welfare. Regarding the impact of government competition on social welfare, Chen et al. (2022) incorporated the structure of central and local governments into a standard environmental dynamic stochastic general equilibrium (E-DSGE) model and found that fiscal decentralization in government fiscal competition is beneficial for improving social welfare and alleviating environmental pollution [38]. In contrast, Hines (2006) argued that government economic competition can lead governments to focus only on economic growth, which can distort fiscal spending, squeeze social spending on, and reduce social welfare [39].

The relationship between government competition and ecological welfare performance. Lv et al. (2022) measured the ecological welfare performance of 30 Chinese provinces and found that government competition has a promoting effect, but there is regional heterogeneity [40]. Lei et al. (2019) explored the impact of local government behavior on carbon welfare performance and found that there is a "complementary relationship" between intergovernmental competition and government size expansion on carbon welfare performance [41]. Wu and Wei (2022) studied the impact of foreign direct investment on local government ecological welfare performance and found that malicious government competition introduced by foreign direct investment is detrimental to improving EWP [42]. However, they also found that the regional distribution and intensity of fiscal revenue and expenditure competition can affect the impact of government's foreign investment competition on EWP, and they believe that the flow of capital, technology, and labor has a significant moderating effect on the impact of foreign investment competition on EWP.

Overall, although some literature has examined the relationship between government competition and ecological welfare performance, the research mainly focuses on the impact of local government economic competition on ecological welfare performance, lacking a multidimensional perspective. Since the release of the "Notice on Improving the Performance Assessment of Local Party and Government Leading Teams and Leading Cadres" by the Central Organization Department of China in 2013, which requires the inclusion of environmental protection in performance assessments, both government competition behavior and ecological welfare performance have undergone corresponding changes. However, existing literature has not fully recognized the potential impact of changes in performance assessments on ecological welfare performance. Against this backdrop, this study makes three contributions. First, it compares the impact of government competition on ecological welfare performance under economic performance assessment and environmental protection assessment from theoretical and empirical perspectives, discussing its short-term, long-term, and spillover effects. Second, considering the limitations of a single assessment system, this study incorporates economic performance assessment and environmental protection assessment into a multidimensional performance assessment system and explores the optimal ratio to promote improvements in ecological welfare performance. The results of this study provide empirical support and decision-making reference for the formulation of assessment systems and the improvement of ecological welfare performance. Third, in terms of spillover effects, this study conducts an in-depth investigation. Current scholars mainly focus on the spillover effects of government competition under economic performance assessments, but this study also examines the spillover effects of government competition under environmental protection assessments. This research

is of great significance for understanding the spillover effects of government competition and provides guidance for regional cooperation and coordination.

## 3 Theoretical analysis and research hypothesis

The government's competitive behavior is mainly constrained and influenced by the officials' appraisal system. Based on relevant policy documents such as the *Decision of the Central Committee of the Communist Party of China on Some Major Issues Concerning Comprehensively Deepening the Reform*, this paper argues that economic development and ecological protection account for a large proportion of the officials' appraisal system, so this paper focuses on the theoretical mechanism of government competition on EWP under economic performance appraisal and environmental performance appraisal.

### 3.1 Direct effects of government competition affecting EWP

In the vertical management system of "centralized political power and decentralized economic power", local government officials act as agents of the central government to assist their principals in achieving national strategic goals, and the central government also decides whether to promote local government officials based on their performance [7, 43]. Therefore, the competitive behavior of local governments is mainly governed by the official appraisal system. Prior to 2013, economic development was the central goal of the Chinese government, and GDP growth became an important indicator for the appraisal of officials. Under the economic performance appraisal, it was difficult for local governments to stimulate economic growth by stimulating consumer demand, and promoting investment became an option for government competition [44]. In order to win economic competition, local governments implement a series of measures such as preferential subsidies and low tax policies to encourage investment in heavy industrial enterprises with political star effect and relax regulations on the entry of highly polluting and energy-consuming enterprises. These governmental behaviors encourage enterprises to over-exploit natural resources and to inject quantity without treating pollution, and even governments will compete to lower the level of environmental regulations in order to compete for the entry of heavy industry enterprises, further triggering the "bottom-up effect" of environmental regulations among regions [45], which is not conducive to the rational opening of resources and the protection of ecological environment. In addition, a well-developed infrastructure can improve the productivity of enterprises. Local officials are keen on infrastructure development to attract factor inflows, crowding out public spending on education and health care and negatively treating social security and social services in their jurisdictions [46]. Economic competition from the government also leads to the capitalization of land, where the government sells land to increase fiscal revenue and strengthen economic competitiveness, driving up property prices, which is detrimental to the well-being of residents and the development of EWP [47].

Local government officials in China have short terms of office, generally 3–5 years each, and economic performance appraisals can catalyze short-sighted behavior of local governments. Local officials use administrative means to interfere with the free flow of factors in order to achieve better economic results during their tenure. While raising the barriers to entry for foreign firms and goods, local governments also support local firms through subsidies and lower tax rates, forming a soft budget constraint protectionism [48], which is detrimental to the allocation efficiency of resources and the layout of industries. Public services are characterized by long investment cycles and slow returns, making them difficult to be the optimal decision for local governments to beautify their performance. Local governments will prefer heavy industries that can boost GDP growth in the short term, resulting in insufficient supply

of public services and ecological damage in the long run [49]. Under economic assessment, the short-sighted behavior of the government can worsen the "resource curse" effect. The large rents of resource-based economies create a breeding ground for rent-seeking, and government competition increases approvals for resource exploitation, which pre-empts the future benefits of resource-based economies [50], leading to the unsustainability of resource-based economies and negatively affecting long-term EWP.

The negative effects of economic competition on current and long-term EWP will be mitigated when the proportion of environmental assessment in the performance assessment system of officials is increased, i.e., a multidimensional assessment system is formed. In order to avoid being "vetoed" due to environmental issues, local governments will, on the one hand, promote the transformation of environmental regulation from "bottom-up competition" to "top-up competition", increase the financial investment in green innovation, and encourage enterprises' green technology research [51]. At the same time, since residents have the right to "vote with their feet", ecological damage in their jurisdictions may cause them to move to other areas. In order to avoid population loss and social stability, the government will further strengthen environmental regulations from "bottom-up competition" to "top-down competition" [52], which is conducive to the improvement of EWP in the short term. On the other hand, local governments will use low taxes, subsidies, and relaxed financing constraints to attract high-quality talent and FDI, and guide high-technology firms to build local factories, generating industrial cluster effects and knowledge spillovers and promoting the jurisdiction's innovation capacity [53]. Therefore, government competition under multidimensional assessment can improve the EWP in the long term. Accordingly, Hypothesis 1a and Hypothesis 1b are proposed in this paper.

**Hypothesis 1a**: Economic competition inhibit EWP in the current period as well as in the long period.

**Hypothesis 1b**: The negative effect of economic competition on EWP is mitigated both in the current and long term when a multidimensional assessment is formed.

## 3.2 Spatial spillover effects of government competition affecting EWP

Governmental competition is not a decision-making behavior of individual local governments. Local governments will achieve their goals by imitating and playing games with neighboring regions, and there are strategic interactions among governments [54]. In other words, government competition can also have an impact on the EWP of neighboring areas.

Government competition under economic performance appraisal leads to negative spatial spillover, where local governments try to improve local EWP at the expense of the EWP of neighboring areas, resulting in "cut-throat competition". Local governments under economic assessment fall into the dominant strategy of no treatment in consideration of the "free-rider" behavior of the surrounding areas towards environmental treatment [55]. Local governments even take advantage of spillover effects to discharge pollutants into neighboring cities, making each other beggarly and mutually disruptive. In addition, the tendency of talent and capital to profit, mobility factors will tend to be in areas with better factor allocation and higher marginal returns, so the increase of local government economic competition will have a siphoning effect on the surrounding talent, capital and technology [56], which is not conducive to the development of EWP in the surrounding areas. At the same time, the concentration of such factors will further strengthen the competitive behavior of the government, the circular accumulation

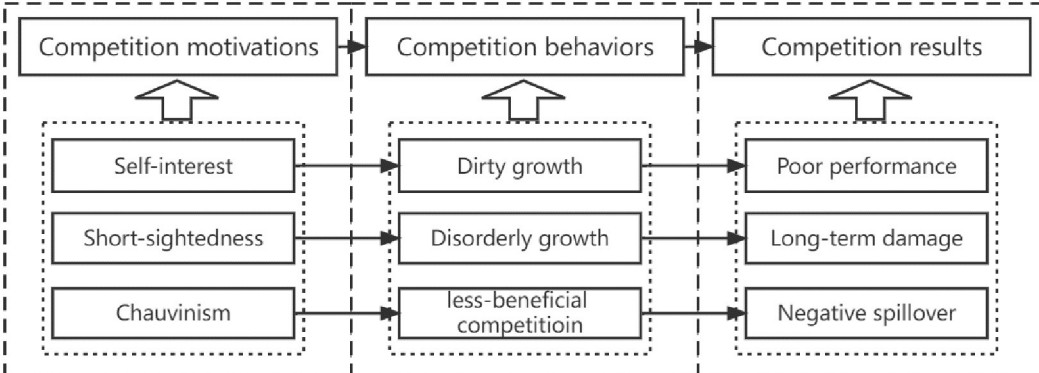

**Fig 1. Logic diagram of government competition affecting EWP under economic performance assessment.**

mechanism will be strengthened, and the ecological welfare performance will finally form a "core-periphery" pattern in the region. Therefore, the enhancement of local economic competition will inhibit the improvement of EWP in the surrounding areas. The detailed logic diagram can be seen in Fig 1.

When a multidimensional performance appraisal system is formed, government officials will place a certain amount of emphasis on the ecological environment for a smooth promotion. The improvement of ecological environment requires joint prevention and control among neighboring regions, thus weakening local protectionism and regional barriers. Neighboring governments compete to learn and imitate advanced social welfare concepts and green and clean technologies, generating knowledge spillovers that improve the EWP of neighboring provinces [9]. Competition among local governments also leads to dynamic adjustment of factors between the province and neighboring provinces. The spatial mobility of factors is essentially a Pareto improvement process that avoids misallocation of factors [57]. Therefore, government competition under multidimensional performance assessment will improve the spatial allocation efficiency of factors and promote the improvement of EWP in neighboring provinces. Accordingly, this paper proposes hypotheses 2a, 2b.

**Hypothesis 2a**: Economic competition inhibits the EWP of neighboring areas.

**Hypothesis 2b**: Multidimensional competition mitigates the negative effects of economic competition on the EWP of neighboring regions.

## 4 Methodology and data

### 4.1 Super-SBM model with undesirable outputs

Since the traditional SBM model takes the DMUs with efficiency figures which larger than 1 as 1, resulting in the inability to distinguish between efficient DMUs, and the SBM model also ignores the problem of non-expected output with negative externalities [58]. To address these issues, the Super-SBM model has been introduced, which integrates the advantages of the super-efficiency model and the SBM model. Compared to the traditional SBM model, the Super-SBM model is able to consider the issue of unexpected outputs in the ecological system, and further differentiate the performance differences among efficient DMUs. Additionally, this model has a certain tolerance for data noise and errors, thus reducing the impact of data

noise on the results [59, 60]. Therefore, the Super-SBM model enables a more comprehensive, accurate, and precise measurement by taking into account various factors related to ecological welfare performance. The model is as follows:

$$\min \rho = \frac{\frac{1}{m} \sum_{i=1}^{m} (\overline{x}/x_{ik})}{\frac{1}{r_1+r_2} \left( \sum_{s=1}^{r_1} \overline{y^d}/y_{sk}^d + \sum_{q=1}^{r_2} \overline{y^u}/y_{qk}^u \right)} \tag{1}$$

$$\begin{cases} \overline{x} \geq \sum_{j=1,\neq k}^{n} x_{ij\lambda_j}, \overline{y^d} \leq \sum_{j=1,\neq k}^{n} y_{Sj}^d \lambda_i \\ \overline{y^d} \geq \sum_{j=1,\neq k}^{n} y_{qj}^d, \overline{x} \geq x_k \\ \overline{y^d} \leq y_k^d, \overline{y^u} \geq y_k^u \\ \lambda_j \geq 0, i = 1,2,..,m, j = 1,2,\ldots,n \\ s = 1,2,\ldots,r_1, q = 1,2,\ldots,r_2 \end{cases} \tag{2}$$

In the equation, ρ is representing the value of EWP in each province, n is the number of DMUs. m, $r_1$, $r_2$ are the input, desired output and non-desired output of each DMU respectively. x, $y^d$, $y^u$ correspond to the elements of the input matrix, desired output matrix and non-desired output matrix of DMUs respectively.

## 4.2 Dynamic panel model

The dynamic GMM model has the following advantages compared to other models. Firstly, it can address endogeneity issues. By introducing a lagged explanatory variable, the dynamic GMM model is able to effectively handle endogeneity problems. Secondly, the dynamic GMM model is better at capturing the changes and dynamic nature of time series data [61–63]. For example, government competition and ecological welfare performance may vary over time. At the same time, there may be endogeneity issues between government competition and ecological welfare performance [64]. To avoid the aforementioned problems from interfering with the empirical results of this study, a lagged dependent variable is used to construct the dynamic GMM model. The model is formulated as follows.

$$\text{Ewp}_{it} = \alpha_0 + \alpha_1 \text{Ewp}_{it-1} + \alpha_2 X_{it} + \varepsilon_{it} \tag{3}$$

i and t represent province i and year t, respectively, EWP is ecological welfare performance, which is used as the explanatory variable in this paper. X is the set of core explanatory variables and control variables, including government competition under economic assessment, government competition under environmental assessment and each control variable, $\varepsilon_{it}$ is a random disturbance term.

## 4.3 Spatial dynamic panel model

There are obvious strategic interactions and games between local governments and neighboring governments when competing, so it is necessary to discuss the spatial dependence and spatial correlation between government competition and EWP in this paper.

Spatial econometric models can be divided into spatial autocorrelation (SAR), spatial error (SEM) and spatial Durbin (SDM) models. SAR mainly considers whether there is spatial correlation between dependent variables and SEM mainly studies whether there is spatial spillover of omitted variables or random error terms. SDM combines the advantages of SAR and SEM models. In this paper, the SDM model is chosen to consider the spatial correlation of the

explanatory variables, explanatory variables and error terms at the same time. Referring to the study of Bouayad-Agha (2010), this paper constructs a dynamic spatial Durbin GMM model by introducing a one-period lag of EWP into the explanatory variables on the basis of a static Durbin model, which is as follows [65].

$$\text{Ewp}_{it} = \alpha_0 + \rho \sum_{j=1}^{N} W_{ij} ewp_{it} + ewp_{it-1} + \lambda \sum_{j=1}^{N} W_{ij} ewp_{it-1} + \beta X_{it} + \theta \sum_{j=1}^{N} W_{ij} X_{it} + \varepsilon_{it} \quad (4)$$

$W_{ij}$ is the spatial matrix, and this paper is based on the inverse distance matrix for empirical analysis. $\rho$ represents the spatial autocorrelation coefficient, and $\lambda$ is the effect of the EWP of the surrounding area in the previous year on the local EWP in this year. $\varepsilon_{it}$ is the spatial error autocorrelation term, and $X_{it}$ is the explanatory variable of the i province in year t.

### 4.4 Data sources and variable description

**4.4.1 Dependent variables.** EWP refers to the efficiency of converting ecological resource consumption into social welfare, and this paper refers to the current definition of EWP and existing measurement studies [21, 23, 59]to construct a system of indicators reflecting the balanced development of ecological environment and social welfare. Based on the principles of scientificity, systematicity and operability, the article uses the consumption of ecological resources as input indicators, mainly involving energy consumption, water consumption and land consumption. Output indicators are mainly divided into non-desired outputs that express environmental pollution and desired outputs that express social welfare value, while environmental pollution is measured by wastewater discharge, exhaust gas discharge and solid waste discharge. The social welfare value is constructed mainly by referring to the Human Development Index [66], which measures three dimensions: health care, economic development and education. The specific indicator measures are shown in Table 2.

**4.4.2 Core explanatory variable.** Under different performance appraisal systems, government competition manifests itself in different forms.

Government competition under economic performance appraisal. In this paper, referring to Huo et al.'s (2023) classification criteria, government competition is mainly expressed as fiscal competition, investment competition and capital attraction competition [67]. Fiscal competition represents the aggressiveness of a region's independent economy, capital competition shows the degree of local efforts to improve infrastructure, and capital attraction competition represents the ability of local governments to attract liquidity factors. In this paper, referring to the studies of Wu et al. (2020), fiscal competition is calculated as the ratio of fiscal expenditure to fiscal revenue, investment competition is measured by the proportion of local fixed asset

**Table 2. Input-output indicators of EWP.**

| Dimension | Criteria | Two level Index | Indicators |
|---|---|---|---|
| Input indicators | Ecological resource consumption | Energy consumption | Energy consumption per capita |
| | | Water consumption | Water consumption per capita |
| | | Land consumption | Built-up area per capita |
| Output indicators | Social welfare | Health care | Average life expectancy |
| | | Education | Years of schooling per capita |
| | | Economy | GDP per capita |
| | Environmental pollution | Wastewater discharge | Industrial wastewater discharge per capita |
| | | Exhaust gas discharge | Industrial sulfur dioxide emissions per capita |
| | | Solid waste discharge | Industrial solid waste generation per capita |

investment in the country, and capital attraction competition is measured by the proportion of foreign direct investment in the country [68]. Finally, the entropy weight method is used to calculate the government competition index under economic assessment, and the larger the index indicates the stronger the economic competitiveness of local governments.

Government competition under environmental performance assessment. The government's environmental competition is mainly measured by drawing on Yang (2020), and the three indicators of solid waste utilization rate, sulfur dioxide removal rate, and dust removal rate are synthesized into an environmental competition index by the entropy weight method [69].

Government competition under multi-dimensional performance appraisal. Economic competition is not conducive to the protection of ecological environment, and ecological competition is not conducive to economic development and social stability, so how to break the contradiction between economic development and environmental protection has become a key issue in the adjustment of the appraisal system. In this paper, economic competition and environmental competition are integrated into the multidimensional assessment system based on the unification of their quantitative scales and the government competition under multidimensional assessment is formed by dynamic adjustment of the weight of economic and environmental assessment [70]. For example, multidimensional competition(1,9) = economic competition * 0.1 + environmental competition * 0.9, and so on to calculate multidimensional competition(2,8), multidimensional competition(3,7), multidimensional competition(4,6), etc.

**4.4.3 Control variables.** To avoid the influence of omitted variables on the regression results, this study selects the following control variables.

Marketization: In a market economy, the market mechanism guides resource allocation and environmental behavior. Moderate marketization may promote innovation and resource efficiency, but excessive marketization may lead to environmental degradation and overexploitation of natural resources. Marketization data is sourced from scholars such as Fang Gang's "China Marketization Index" [42, 71].

Industrial structure: The industrial structure is closely related to ecological welfare. Different industries have varying degrees of impact on the environment. For example, heavy industries may have negative impacts on the environment, while clean energy industries may contribute positively to the environment. The industrial structure is measured by the ratio of value-added from the tertiary industry to value-added from the secondary industry [72].

Technological development: There is an interaction between technological development and ecological welfare. On the one hand, technological development may bring negative impacts such as environmental pollution and resource consumption. On the other hand, technological development can also provide innovative technologies and solutions to address environmental issues. Technological development is measured by the number of patents per unit of GDP [30].

Economic development: There is a complex relationship between economic development and ecological welfare. On the one hand, economic development may lead to negative impacts such as natural resource depletion and environmental pollution. On the other hand, a higher level of economic development may provide more resources and technology to promote environmental actions. By controlling for economic development factors, research can better explore the balance and coordination between government, economic development, and ecological welfare. Economic development is represented by per capita GDP [27, 73].

**4.4.4 Data source.** This paper selects panel data of 30 provincial-level regions in China (excluding Hong Kong, Macao and Tibet) from 2005–2019 for analysis, and the relevant data are mainly obtained from the website of the National Bureau of Statistics of China, *China Demographic Statistical Yearbook* and *China Education Statistical Yearbook*, and some missing

**Table 3. Descriptive statistics of variables.**

| Variable | Definition | Min | Med | Max | Mean |
|---|---|---|---|---|---|
| Eco | Economic Competition | 0.050 | 0.186 | 0.678 | 0.225 |
| Env | Environmental Competition | 0.076 | 0.308 | 0.748 | 0.310 |
| Mark | Marketization | 2.372 | 6.410 | 11.109 | 6.485 |
| Ind | Industrial Structure | 0.500 | 0.889 | 5.169 | 1.057 |
| Pat | technological development | 0.001 | 0.017 | 7.793 | 0.138 |
| Gdpp | Economic Development | 0.051 | 0.376 | 1.642 | 0.426 |

data are filled in using the interpolation method. The descriptive statistics of each variable are shown in Table 3.

## 5 Empirical results

### 5.1 Government competition under different assessment systems and EWP

**5.1.1 Short-term effect.**   Table 4 reports the results of the systematic GMM regression of the effects of economic and environmental competition on the EWP in the current period. L. EWP is the EWP level of the previous period, and the results of the base regression show that the coefficient of L.EWP is significantly positive, which indicates that if the EWP of the previous period is at a high level, the EWP of the next period will also continue to be high, and the EWP has a strong path-dependent characteristic in the time dimension, i.e., it shows an obvious "snowball" effect and "virtuous circle". Columns (1)-(2) of Table 4 show that the coefficient of government competition under economic assessment (Eco) is negative and passes the 1% level of significance. Under the economic performance assessment, local governments are more likely to sacrifice the ecological environment in exchange for maximizing the economic benefits of their jurisdictions. At the same time, the government also squeezes public service expenditures and erodes residents' welfare, resulting in a reduction of EWP in the current period. Government competition under environmental assessment (Env) can significantly promote EWP, indicating that when local governments fully carry out environmental competition, local governments tend to increase their attention to the environment and improve environmental management capacity to meet the central government's assessment policy, which in turn promotes the improvement of EWP. To ensure the robustness of the regression results, this paper uses replacement of explanatory variables and shortened sample time for regression analysis, respectively. Technology market is the carrier of technology goods trading, which better reflects the transformation and allocation efficiency of technology, etc. Therefore, this paper refers to Lin et al.'s (2021) study and replaces the number of patents per unit GDP (Pat) with the turnover of technology market per unit GDP (Tecm) to measure the level of technology development, and the results are shown in (3)-(4) [74]. After replacing the explanatory variables, the significance as well as the sign of economic competition and environmental competition do not change, so the conclusions of this paper are robust. Since the COVID-19 had a serious impact on the macroeconomy and the data in 2020 were volatile, which may affect the conclusions. This paper excludes the 2020 sample for regression, and the results are shown in (5)-(6), and the conclusions of this paper remain robust after the exclusion of the sample. The GMM model is based on the lagged terms of the explanatory and explanatory variables as instrumental variables for regression, and it is clear from AR(1), AR(2) and Hansen's test that the instrumental variables in this paper are more reasonably set, and thus can overcome the effect of endogeneity on the findings.

**Table 4. The short-term effect under different assessment systems.**

|  | (1) | (2) | (3) | (4) | (5) | (6) |
|---|---|---|---|---|---|---|
|  | EWP | EWP | EWP | EWP | EWP | EWP |
| L.EWP | 0.5633*** | 0.6195*** | 0.4434*** | 0.5749*** | 0.5545*** | 0.5751*** |
|  | (22.43) | (45.47) | (26.73) | (35.51) | (33.90) | (50.15) |
| Eco | -0.3898*** |  | -0.2675*** |  | -0.1788*** |  |
|  | (-3.21) |  | (-3.55) |  | (-2.61) |  |
| Env |  | 0.3985*** |  | 0.6189*** |  | 0.9997*** |
|  |  | (5.27) |  | (11.11) |  | (7.51) |
| Mark | 0.0364*** | 0.0200*** | 0.0371*** | 0.0250*** | 0.0303*** | 0.0442*** |
|  | (12.23) | (16.12) | (14.58) | (28.61) | (13.40) | (19.25) |
| Ind | 0.1282*** | 0.1262*** | 0.0585*** | 0.0309*** | 0.1961*** | 0.1167*** |
|  | (7.80) | (21.75) | (5.24) | (3.62) | (12.03) | (10.45) |
| Pat | -0.0495*** | -0.0120*** |  |  | -0.0513*** | 0.0038 |
|  | (-5.90) | (-2.64) |  |  | (-6.12) | (0.71) |
| Tecm |  |  | 0.0005*** | 0.0003*** |  |  |
|  |  |  | (10.80) | (9.42) |  |  |
| Gdpp | -0.2457*** | -0.3719*** | -0.4129*** | -0.4257*** | -0.3478*** | -0.5654*** |
|  | (-8.18) | (-15.32) | (-23.01) | (-19.18) | (-13.79) | (-14.54) |
| cons | 0.1818*** | 0.0803*** | 0.3313*** | 0.0991*** | 0.1632*** | -0.1342*** |
|  | (7.75) | (3.79) | (12.32) | (4.28) | (6.73) | (-4.49) |
| AR(1) | -2.22 | -2.27 | -2.18 | -2.25 | -2.45 | -2.41 |
|  | [0.026] | [0.023] | [0.029] | [0.024] | [0.014] | [0.016] |
| AR(2) | 1.23 | 1.48 | 1.38 | 1.59 | 1.45 | 1.59 |
|  | [0.217] | [0.138] | [0.168] | [0.113] | [0.146] | [0.111] |
| Hansen test | 22.54 | 25.7 | 25.11 | 25.09 | 26.32 | 23.85 |
|  | [0.99] | [0.814] | [0.996] | [0.837] | [0.979] | [1.00] |
| N | 420 | 420 | 420 | 420 | 390 | 390 |

Note:

***, **, and * indicate significance at 1%, 5%, and 10% levels, respectively.

Figures in () are the standard errors, and those in [] are the p-values of the corresponding test statistics.

**5.1.2 Long-term effect.** Verifying whether the government's competitive behavior is short-sighted is important for the development of the appraisal system. In this paper, the EWP from the next 1th year to the next 4th years is regressed by the system GMM, and the regression results are shown in Table 5. From the results of the regression coefficients of Eco, the government competition under economic assessment inhibits the EWP in the future, which verifies hypothesis 1a. Further, the negative effect will weaken in the next 1th and next 2th years, from -0.39 in the current period to -0.32 in the next 1th year and -0.11 in the next 2th years, but the negative effect will be enhanced in the next 3th and next 4th years. This indicates that economic competition can achieve rapid economic development and drive social welfare in the near one or two years, and thus the inhibitory effect on EWP will be weakened, but the negative effect of economic competition will be further enhanced in the far future with the deterioration of the environment and the lack of innovation capacity. From the regression results of Env, the positive effect of government competition on EWP under environmental assessment also has a trend of first weakening and then increasing, from 0.40 in the current period to 0.22 in the coming 1th year and 0.14 in the coming 2th years, and then increasing to

**Table 5. The Long-term effect under different assessment systems.**

|  | (1) | (2) | (3) | (4) | (5) | (6) | (7) | (8) |
|---|---|---|---|---|---|---|---|---|
|  | Future 1 EWP | | Future 2 EWP | | Future 3 EWP | | Future 4 EWP | |
| L.y | 0.5258*** | 0.8052*** | 0.5254*** | 0.8009*** | 0.5218*** | 0.8016*** | 0.4342*** | 0.7543*** |
|  | (31.21) | (68.30) | (38.82) | (67.04) | (42.96) | (76.06) | (39.33) | (84.91) |
| Eco | -0.3183*** |  | -0.1101** |  | -0.5954*** |  | -0.6148*** |  |
|  | (-3.66) |  | (-2.29) |  | (-6.70) |  | (-8.93) |  |
| Env |  | 0.2176*** |  | 0.1363*** |  | 0.2044*** |  | 0.2575*** |
|  |  | (6.28) |  | (3.16) |  | (5.66) |  | (11.16) |
| Control variables | YES | YES | YES | YES | YES | YES | YES | YES |
| AR(1) | -2.3 | -2.21 | -2.27 | -2.32 | -2.45 | -2.27 | -2.14 | -2.21 |
|  | [0.022] | [0.027] | [0.023] | [0.020] | [0.014] | [0.023] | [0.033] | [0.027] |
| AR(2) | 1.26 | 1.31 | 1.23 | 1.44 | 1.49 | 1.38 | 1.27 | 1.22 |
|  | [0.208] | [0.192] | [0.218] | [0.149] | [0.136] | [0.168] | [0.203] | [0.222] |
| Hansen test | 24.46 | 26.48 | 25.87 | 27.04 | 19.77 | 25.53 | 22.79 | 26.03 |
|  | [0.956] | [0.600] | [0.869] | [0.570] | [0.955] | [0.650] | [0.786] | [0.571] |
| N | 420 | 420 | 390 | 390 | 360 | 360 | 330 | 330 |

0.26 in the next 4th years. The development of environmental competition will increase the government's requirement for firms to green their production, and firms will invest more in pollution control and cleaner production, thus eroding their economic output [75]. Therefore, environmental competition will lead to a reduction in social welfare within the next year or two, which will partially offset the positive effects overall. However, in the long run, it takes time for the effects of green technology development to emerge, so environmental competition creates an "innovation compensation effect" in the long run [76], and its positive effects are increasing in the long run.

**5.1.3 Spatial spillover effect.** The competition among local governments has obvious game and strategy interaction problems. In this paper, we next use the spatial GMM model to study the impact of government competition on EWP can make up for the lack of dependent on geographical location, and the regression results are shown in Table 6. Columns (1)-(2) of Table 6 show the regression results of the spatial GMM based on the inverse distance matrix (w1). The regression results of ρ are significantly positive, i.e., the improvement of EWP in the surrounding areas can promote the local EWP, and there is spatiality in the development of EWP. The results in column (1) show that the regression coefficient of Eco is negative and passes the 5% significance test, further verifying the previous conclusion. w*Eco is significantly negative at the 1% level, and the increase in economic competitiveness of neighboring governments will have a siphoning effect on local talents and capital, and also cause the deterioration of the local ecological environment, thus reducing the level of local EWP. Therefore, economic competition among governments will produce negative spatial spillover effects, and governments will compete with each other and engage in white-hot "cut-throat competition", which verifies hypothesis 2a. Similarly, the regression result of Env in column (2) of Table 6 is significantly positive, and government competition under environmental assessment can promote the EWP, which is consistent with the previous findings. The coefficient of W*Env is positive but does not pass the 10% significance test, indicating that environmental competition in the surrounding areas did not have an impact on local EWP. To ensure the robustness of the empirical results, columns (3)-(4) of Table 6 report the regression results after transforming from the inverse distance matrix (w1) to the economic matrix (w2). The resulting coefficients

**Table 6. The spatial spillover effect under different assessment systems.**

| | (1) | (2) | (3) | (4) |
|---|---|---|---|---|
| | EWP (w1) | | EWP (w2) | |
| L.y | 0.5144*** | 0.3306*** | 0.4532*** | 0.7391*** |
| | (12.36) | (4.03) | (7.22) | (21.60) |
| | (-3.74) | (-3.15) | (-1.71) | (-1.52) |
| Eco | -0.3829** | | -0.5146** | |
| | (-2.19) | | (-2.29) | |
| Env | | 1.1335*** | | 0.5661*** |
| | | (2.81) | | (2.99) |
| W*Eco | -23.2079*** | | -26.5476* | |
| | (-2.96) | | (-1.65) | |
| W*Env | | 15.3457 | | 17.8992 |
| | | (0.78) | | (1.41) |
| ρ | 33.6337*** | 28.5576*** | 42.8068*** | 26.7169*** |
| | (8.33) | (7.11) | (5.17) | (2.69) |
| Control variables | YES | YES | YES | YES |
| AR(1) | -2.14 | -2.13 | -2.16 | -2.28 |
| | [0.033] | [0.033] | [0.030] | [0.022] |
| AR(2) | 1.33 | 1.58 | 1.11 | 1.49 |
| | [0.183] | [0.114] | [0.267] | [0.136] |
| Hansen test | 13.67 | 13.98 | 16.55 | 19.38 |
| | [0.954] | [0.928] | [0.957] | [0.886] |
| N | 420 | 420 | 420 | 420 |

exhibit consistency in sign and significance, and the empirical results of spatial GMM are robust. Based on the above analysis, how to circumvent the negative spillover effect of the existence of economic competition through the adjustment of the appraisal system becomes an urgent problem to be solved.

## 5.2 Government competition under multidimensional assessment systems and EWP

The Chinese government has been focusing on economic construction since its reform and opening up, and economic growth has become an important indicator for assessing officials. The analysis above shows that government competition under economic assessment not only inhibits EWP in the current and long term, but also negatively affects the EWP of the surrounding areas. However, the complete abandonment of economic assessment in favor of environmental assessment has negative implications for the improvement of people's lives and the long-term stability of society. Since 2013, the Chinese government has gradually increased the proportion of environmental assessment in the officials' system [77], deciphering the conflicts between economic development and environmental protection, immediate and long-term interests, and local and overall interests through the combination of multiple indicators. This paper then seeks to find the optimal assessment weights to achieve EWP growth through dynamic adjustment of multidimensional competition.

**5.2.1 Short-term and long-term effect.** Table 7 shows the effects of government competition on current and forward EWP after the appraisal system has been dynamically adjusted. Multcom19 is a multidimensional government competition with 10% economic assessment

**Table 7. The short-term and long-term effect under multidimensional assessment systems.**

|  | (1) | (2) | (3) | (4) | (5) |
|---|---|---|---|---|---|
|  | Current year EWP | Future 1 EWP | Future 2 EWP | Future 3 EWP | Future 4 EWP |
| Multcom(1,9) | 0.3235*** | 0.3092*** | 0.1774*** | 0.0413 | -0.3230*** |
|  | (5.21) | (8.20) | (3.14) | (0.97) | (-3.90) |
| Multcom(2,8) | 0.3621*** | 0.2712*** | 0.1352* | -0.0717 | -0.4456*** |
|  | (5.13) | (4.34) | (1.85) | (-0.91) | (-6.63) |
| Multcom(3,7) | 0.3396*** | 0.1631** | 0.1177* | -0.1405* | -0.5459*** |
|  | (4.67) | (2.17) | (1.77) | (-1.74) | (-9.17) |
| Multcom(4,6) | 0.2712*** | 0.0408 | 0.0765 | -0.2282*** | -0.6575*** |
|  | (3.89) | (0.56) | (1.19) | (-3.03) | (-11.34) |
| Multcom(5,5) | 0.1954*** | -0.1038* | 0.0156 | -0.3441*** | -0.7469*** |
|  | (3.19) | (-1.68) | (0.28) | (-5.06) | (-11.55) |
| Multcom(6,4) | 0.1203** | -0.2132*** | -0.0464 | -0.4154*** | -0.7835*** |
|  | (2.29) | (-4.08) | (-0.97) | (-6.10) | (-12.23) |
| Multcom(7,3) | 0.0825** | -0.2821*** | -0.0945** | -0.4181*** | -0.6078*** |
|  | (1.99) | (-6.16) | (-2.22) | (-6.68) | (-7.05) |
| Multcom(8,2) | 0.0072 | -0.3524*** | -0.1227*** | -0.4101*** | -0.4825*** |
|  | (0.20) | (-7.00) | (-3.21) | (-7.62) | (-6.69) |
| Multcom(9,1) | -0.0399 | -0.3620*** | -0.2011*** | -0.3852*** | -0.3923*** |
|  | (-1.33) | (-8.00) | (-4.73) | (-8.53) | (-6.87) |

Note: The results are regressions of multidimensional competition for EWP under different weights, respectively. Each GMM regression is put into control variables. With consistent instrumental variables and lag orders, AR (1) p-values are less than 0.1, and AR (2) and Hansen test p-values are greater than 0.1, and the specific test results can be found in the S1 Appendix.

and 90% environmental assessment, similarly Multcom28 is a multidimensional government competition with 20% economic assessment and 80% environmental assessment, and so on. From column (1), it can be seen that when a multidimensional appraisal system is formed, multidimensional competition can mitigate the negative effect of economic competition on current EWP. When the ratio of economic and environmental assessment is between 1:9 and 7:3, multidimensional competition can promote the EWP in the current year. The results in columns (4)-(5) show that even with the formation of a multiple appraisal system, government competition still inhibits the improvement of EWP in the third and fourth years of the future, but the negative effect is diminished compared to economic appraisal. This suggests that government competition with integrated economic and environmental assessment objectives has some advantages over traditional government competition that focuses only on economic growth. Further analysis from columns (2)-(3) shows that government competition under multidimensional appraisal can achieve the development of EWP in the next 2 years when the appraisal ratio is between 1:9 and 3:7. The above analysis fully illustrates that the negative effect of economic competition on EWP is mitigated both in the current and long term when a multidimensional assessment is formed, verifying hypothesis 1b.

**5.2.2 Spatial spillover effect.** Government competition under economic appraisal causes significant negative externalities. this paper further adjusts the appraisal system dynamically to explore whether government competition under optimal weighting of economic appraisal and environmental appraisal can crack the problem of conflicting local and overall interests. the regression results are shown in Table 8. The significance of W*x failed the 10% test under any multidimensional competition, indicating that government competition in the surrounding

**Table 8. The spatial spillover effect under multidimensional assessment systems.**

|  | (1) | (2) | (3) | (4) | (5) | (6) | (7) |
|---|---|---|---|---|---|---|---|
|  | L.y | x | W*x | ρ | AR(1) | AR(2) | Hansen test |
| Multcom(1,9) | 0.3689*** | 1.3953* | -5.9039 | 26.7634*** | -2.01 | 1.38 | 17.49 |
|  | (4.00) | (1.85) | (-0.20) | (5.10) | [0.045] | [0.167] | [0.863] |
| Multcom(2,8) | 0.5177*** | 2.0482* | -35.1070 | 21.1383*** | -2.60 | 1.34 | 13.09 |
|  | (7.25) | (1.77) | (-0.81) | (4.98) | [0.009] | [0.182] | [0.975] |
| Multcom(3,7) | 0.5196*** | 0.5607 | -10.9636 | 17.3453*** | -2.23 | 1.51 | 15.45 |
|  | (6.47) | (0.56) | (-0.28) | (3.48) | [0.026] | [0.130] | [0.930] |
| Multcom(4,6) | 0.6124*** | -0.9221** | 11.2236 | 22.7424*** | -1.99 | 1.33 | 13.58 |
|  | (7.32) | (-2.36) | (0.29) | (4.20) | [0.047] | [0.185] | [0.969] |
| Multcom(5,5) | 0.5206*** | -0.9583*** | -22.1865 | 27.5406*** | -1.86 | 1.20 | 13.84 |
|  | (8.08) | (-3.46) | (-0.54) | (5.60) | [0.063] | [0.230] | [0.964] |
| Multcom(6,4) | 0.5293*** | -0.9216*** | -38.8862 | 29.9681*** | -1.83 | 1.21 | 12.4 |
|  | (8.94) | (-3.52) | (-1.29) | (6.57) | [0.067] | [0.225] | [0.983] |
| Multcom(7,3) | 0.5375*** | -0.8459*** | -21.9762 | 29.2674*** | -1.89 | 1.31 | 13.12 |
|  | (8.80) | (-2.91) | (-1.31) | (7.21) | [0.059] | [0.191] | [0.975] |
| Multcom(8,2) | 0.4744*** | -0.5296** | -7.4681 | 23.3818*** | -1.93 | 1.34 | 16.86 |
|  | (7.06) | (-1.96) | (-0.40) | (4.14) | [0.053] | [0.179] | [0.887] |
| Multcom(9,1) | 0.4724*** | -0.4831** | -9.2404 | 24.1089*** | -1.93 | 1.35 | 17.31 |
|  | (6.82) | (-2.01) | (-0.56) | (4.26) | [0.054] | [0.176] | [0.870] |

areas did not affect local EWP under multidimensional performance appraisal. Meanwhile, when the weight of economic and environmental assessment is 1:9–2:8, the regression coefficient of x is significantly positive, i.e., multidimensional competition can promote local EWP without affecting the surrounding areas. Multidimensional competition ameliorates the one-loss problem that economic competition that is both detrimental to local and surrounding EWP, verifying hypothesis 2b.

## 6 Conclusions and policy implications

In the context of sustainable development, the central government has been continuously adjusting the assessment system for officials in order to achieve high-quality economic development. This study uses panel data samples and employs the super-efficiency SBM (Undesirable Outputs) to calculate the ecological welfare performance, exploring the impact of the assessment system on government competition and ecological welfare performance. Our research findings are as follows: ①Different government assessment systems have different effects on government competition and ecological welfare performance. Under the economic assessment system, government competition may have a negative impact on current and future ecological welfare performance, while the environmental assessment system can promote the improvement of ecological welfare performance. ② The spillover effects of government competition have significant differences in the impact on the ecological welfare performance of neighboring governments. Environmental competition in neighboring regions enhances local ecological welfare performance, while economic competition has a negative impact on local ecological welfare performance. ③ A multidimensional performance assessment system helps to alleviate the contradictions between government's economic development and environmental protection, short-term and long-term interests, and local and overall interests, and promotes the improvement of ecological welfare performance. After dynamically setting the weight of the

assessment, government competition can promote the current and future improvement of ecological welfare performance. When the weight of economic and environmental assessments is in the range of 1:9 to 7:3, government competition contributes to the improvement of current ecological welfare performance. When the weight is in the range of 1:9 to 3:7, government competition contributes to the improvement of future ecological welfare performance.④ Under the multidimensional performance assessment, the spillover effects of government competition are not significant, and the competition among neighboring governments does not affect local ecological welfare performance. At the same time, multidimensional competition can promote the improvement of local ecological welfare performance, solving the problems of economic competition inhibiting the growth of neighboring ecological welfare performance and its negative impact on local ecological welfare performance.

Based on the results of the empirical evidence, this paper offers some policy implications as follows.

First, introduce multidimensional performance assessment and integrate environmental assessment into the assessment system. When evaluating government performance, the balance between economy and ecology should be fully considered, focusing on long-term development and shared social value. Establish a scientific, fair, and actionable system of assessment indicators to ensure the scientific and fair assessment of government performance. Establish a clear and coordinated assessment mechanism to ensure the coordinated cooperation of various departments in fulfilling their environmental protection responsibilities and tasks.

Second, promote intergovernmental regional coordination and cooperation, establish regional cooperation mechanisms, strengthen multidimensional competition, and promote the improvement of local ecological welfare performance. The government needs to develop ecological environment protection plans, scientifically allocate environmental resources, increase investment in ecological environment protection, and achieve sustainable development. At the same time, efforts should be made to reduce the spillover effects of government competition to prevent the negative impact of economic competition among neighboring regional governments on local ecological welfare performance.

Third, in policy implementation, strengthen regulatory efforts, and pay attention to the construction of supporting assessment and incentive measures to ensure the effective implementation of policies. The government and enterprises should work together to establish a good cooperation model for environmental protection and sustainable development, promote the sustainable development of the economy and society, including public education, information disclosure, technological innovation, and environmental risk management. At the same time, the government needs to implement long-term, coordinated, and integrated policies to ensure the long-term and stable improvement of ecological welfare performance.

## Supporting information

**S1 Appendix.**
(DOCX)

**S1 Data.**
(ZIP)

## Author Contributions

**Conceptualization:** Tianzheng Fan.

**Data curation:** Yanqin Lv, Tianzheng Fan.

**Formal analysis:** Yanqin Lv, Tianzheng Fan.

**Funding acquisition:** Shanhong Li, Yanqin Lv, Tianzheng Fan.

**Investigation:** Shanhong Li, Tianzheng Fan.

**Methodology:** Shanhong Li, Tianzheng Fan.

**Resources:** Shanhong Li, Yanqin Lv, Ziye Zhang.

**Software:** Shanhong Li, Yanqin Lv, Ziye Zhang, Chen Jing.

**Supervision:** Shanhong Li, Ziye Zhang, Chen Jing.

**Validation:** Ziye Zhang, Gao Feng, Chen Jing.

**Visualization:** Gao Feng, Chen Jing.

**Writing – original draft:** Shanhong Li, Gao Feng, Chen Jing.

**Writing – review & editing:** Gao Feng, Chen Jing.

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
