## [Decision Letter · Decision Letter 0]

20 Jun 2023

PONE-D-23-07357Multidimensional Performance Assessment, Government Competition and Ecological Welfare PerformancePLOS ONE

Dear Dr. Yanqin,

Thank you for submitting your manuscript to PLOS ONE. After careful consideration, we feel that it has merit but does not fully meet PLOS ONE’s publication criteria as it currently stands. Therefore, we invite you to submit a revised version of the manuscript that addresses the points raised during the review process.

We look forward to receiving your revised manuscript.

Kind regards,

Umer Shahzad, PhD

Academic Editor

PLOS ONE

Journal Requirements:

"This research is supported by the social science foundation of Xinjiang Autonomous Region, Project No.: 21bjy046; Cultivation project of School of economics and management, Xinjiang University, Project No.: 17jgpy013"

"This research is supported by the social science foundation of Xinjiang Autonomous Region, Project No.: 21bjy046; Cultivation project of School of economics and management, Xinjiang University, Project No.: 17jgpy013"

"NO authors have competing interests"

7. PLOS requires an ORCID iD for the corresponding author in Editorial Manager on papers submitted after December 6th, 2016. Please ensure that you have an ORCID iD and that it is validated in Editorial Manager. To do this, go to ‘Update my Information’ (in the upper left-hand corner of the main menu), and click on the Fetch/Validate link next to the ORCID field. This will take you to the ORCID site and allow you to create a new iD or authenticate a pre-existing iD in Editorial Manager. Please see the following video for instructions on linking an ORCID iD to your Editorial Manager account: https://www.youtube.com/watch?v=_xcclfuvtxQ

8. We note that Figure 2 in your submission contain map/satellite image which may be copyrighted. All PLOS content is published under the Creative Commons Attribution License (CC BY 4.0), which means that the manuscript, images, and Supporting Information files will be freely available online, and any third party is permitted to access, download, copy, distribute, and use these materials in any way, even commercially, with proper attribution. For these reasons, we cannot publish previously copyrighted maps or satellite images created using proprietary data, such as Google software (Google Maps, Street View, and Earth). For more information, see our copyright guidelines: http://journals.plos.org/plosone/s/licenses-and-copyright.

Reviewers' comments:

Reviewer's Responses to Questions

**Comments to the Author**

1. Is the manuscript technically sound, and do the data support the conclusions?

Reviewer #1: Yes

Reviewer #2: Yes

2. Has the statistical analysis been performed appropriately and rigorously? 

Reviewer #1: Yes

Reviewer #2: Yes

3. Have the authors made all data underlying the findings in their manuscript fully available?

Reviewer #1: Yes

Reviewer #2: Yes

4. Is the manuscript presented in an intelligible fashion and written in standard English?

Reviewer #1: Yes

Reviewer #2: Yes

5. Review Comments to the Author

Reviewer #1: This paper studies the influencing factors of ecological welfare performance from the perspective of multi-dimensional government competition. Overall, this topic is interesting and fully contributes to new knowledge. Therefore, I suggest accepting it through appropriate revisions. There are some suggestions that need to be improved on this paper.

1.In the literature review section, there has been a lot of research on the relationship between government competition and ecological welfare performance, but this article has not reflected it, resulting in a weak foundation for the paper. It is recommended to further supplement and improve.

2.Is the usage of the term 'ecological assessment' accurate? The existing literature mostly uses "environmental assessment".

3. Insufficient extraction of innovative points, please further modify and improve them.

4. The variable names "Multi dimensional Competition 19, Multi dimensional Competition 28..." in Table 6 are not standardized and can easily cause ambiguity. It is recommended to modify and improve them.

5. Language needs to be modified and proofread.

Reviewer #2: The contradictions between current and long-term, economic development and environmental protection, and local and surrounding areas are described, and a multidimensional performance assessment system is constructed, which further expands the perspective of ecological welfare performance research. However, the following areas for improvement still exist:

1. The marginal contribution of this paper needs further refinement, for example, marginal contribution1, it seems that there are already studies such as ”Sustainable development of China's regions from the perspective of ecological welfare performance: analysis based on GM(1,1) and the malmquist index”, accounting for environmental pollutants as non-expected output EWP.

2. The literature in the literature review can be updated to the latest year, as appropriate, such as ”Ecological welfare performance, industrial agglomeration and technological innovation: an empirical study based on Beijing–Tianjin–Hebei, Yangtze River Delta and Pearl River Delta””Can New Urbanization Construction Improve Ecological Welfare Performance in the Yangtze River Economic Belt?”, etc.

3. Why choose Super-SBM model and dynamic GMM model? The advantages of these models need further explanation.

4. The paper mentions "Under the economic appraisal in China, government competition is mainly manifested in three aspects: fiscal competition, investment competition and investment attraction competition", is there any basis for this? What is the basis for choosing the four control variables of marketization and industry structure?

5. The conclusion and the abstract are two different sections, so it is recommended to summarize the core content of the conclusion section.

6. The policy recommendations are linked to the conclusions. However, the current policy recommendations seem to emphasize more on how to promote multidimensional performance appraisal, which is clearly not closely linked to the conclusions.

6. PLOS authors have the option to publish the peer review history of their article (what does this mean?). If published, this will include your full peer review and any attached files.

Reviewer #1: No

Reviewer #2: No

---

## [Author Response · Author response to Decision Letter 0]

20 Jul 2023

Manuscript. Number.: PONE-D-23-07357

Title: Multidimensional Performance Assessment, Government Competition and Ecological Welfare Performance

Responses to reviewers' comments on the manuscript PONE-D-23-07357

Dear Editor,

Thanks for the reviewers’ comments and suggestions on our manuscript. We have modified the manuscript accordingly, and the main changes are highlighted with red letters in the revised manuscript. A point-to-point reply is listed below where the comments of reviewers are in black and the replies are in blue.

Reviewer #1:

1.In the literature review section, there has been a lot of research on the relationship between government competition and ecological welfare performance, but this article has not reflected it, resulting in a weak foundation for the paper. It is recommended to further supplement and improve.

Following the reviewer's suggestion, the literature review section has been expanded based on the research on the relationship between government competition and ecological welfare performance to include the following content: "(3) Interregional government competition has a significant impact on ecological welfare performance. .............................................".

2.Is the usage of the term 'ecological assessment' accurate? The existing literature mostly uses "environmental assessment".

Based on the reviewer's suggestion, we have replaced all instances of "ecological assessment" with "environmental assessment" throughout the paper.

3. Insufficient extraction of innovative points, please further modify and improve them.

Based on the reviewer's suggestion, we have made modifications to the innovative aspects and added a new innovative point: "Deepening the study on spillover effects. While most scholars have focused on the spillover effects of government competition under economic performance assessment, this paper also investigates the spillover effects of government competition under environmental assessment. This study contributes to our understanding of the impact of government competition spillover effects and provides guidance for regional cooperation and coordination."

 4. The variable names "Multi dimensional Competition 19, Multi dimensional Competition 28..." in Table 6 are not standardized and can easily cause ambiguity. It is recommended to modify and improve them.

Based on the reviewer's suggestion, we have made modifications to the expression of multidimensional competition in the paper. We have replaced all instances of "multidimensional competition 19, multidimensional competition 28..." with "multidimensional competition (1,9), multidimensional competition (2,8)..."

5. Language needs to be modified and proofread.

According to the reviewer's suggestion, the language has been revised and proofread.

Reviewer #2:

1. The marginal contribution of this paper needs further refinement, for example, marginal contribution1, it seems that there are already studies such as ”Sustainable development of China's regions from the perspective of ecological welfare performance: analysis based on GM(1,1) and the malmquist index”, accounting for environmental pollutants as non-expected output EWP.

 Based on the reviewer's suggestion, we have deleted the marginal contribution I "Counting environmental pollutants as unintended outputs EWP". At the same time, we added an innovation point: "The research is deeply innovative in terms of spillover effects. While contemporary scholars have mainly studied the spillover effects of government competition under economic performance appraisal, this paper also studies the spillover effects of government competition under environmental appraisal, and this study contributes to understanding the impact of government competition spillover effects and also provides guidance for inter-regional cooperation and coordination."

2. The literature in the literature review can be updated to the latest year, as appropriate, such as ”Ecological welfare performance, industrial agglomeration and technological innovation: an empirical study based on Beijing–Tianjin–Hebei, Yangtze River Delta and Pearl River Delta””Can New Urbanization Construction Improve Ecological Welfare Performance in the Yangtze River Economic Belt?”, etc.

Based on the reviewers' suggestions, we updated the literature review section to the latest year, and added all the literature listed by the reviewers to the literature review section.

3. Why choose Super-SBM model and dynamic GMM model? The advantages of these models need further explanation.

Based on the reviewer's suggestion, the paper has added explanations for the advantages of the Super-SBM model and dynamic GMM model, respectively.

Advantages of the Super-SBM model: The Super-SBM model combines the strengths of the super-efficiency model and the SBM model. Compared to the traditional SBM model, the Super-SBM model can address the issue of unexpected outputs in the ecosystem and further differentiate performance differences among efficient DMUs. Additionally, this model has a certain tolerance for data noise and errors, reducing the impact of data noise on the results (Bian et al., 2020; Zhang et al., 2021). Therefore, the Super-SBM model allows for a more comprehensive, accurate, and precise measurement by taking into account various factors related to ecological welfare performance.

Advantages of the dynamic GMM model: The dynamic GMM model boasts several advantages over other models. Firstly, it can effectively handle endogeneity issues. By introducing a lagged explanatory variable, the dynamic GMM model can address endogeneity problems in a more robust manner. Secondly, the dynamic GMM model is better equipped to capture changes and the dynamic nature of time series data (Holtz-Eakin et al., 1990; Arellano and Bond, 1991; Arellano and Bover, 1995). 

4. The paper mentions "Under the economic appraisal in China, government competition is mainly manifested in three aspects: fiscal competition, investment competition and investment attraction competition", is there any basis for this? What is the basis for choosing the four control variables of marketization and industry structure?

Based on the reviewer's suggestion, we have added the rationale for the selection of core explanatory variables and control variables. The statement "government competition is mainly manifested in three aspects: fiscal competition, investment competition, and investment attraction competition" is primarily based on Huo et al. (2023) publication titled "The impact of fiscal decentralization and intergovernmental competition on the local government debt risk: Evidence from China".

The selection of control variables is based on the following:

Marketization: In a market economy, market mechanisms guide resource allocation and environmental behavior. Moderate marketization may promote innovation and resource efficiency, while excessive marketization may lead to environmental degradation and excessive exploitation of natural resources. Data on marketization is sourced from the "China Marketization Index" by scholars such as Fan, G. et al. (Lai et al., 2023; Wu and Dong).

Industrial structure: The industrial structure is closely related to ecological welfare. Different industries have varying degrees of impact on the environment. For example, heavy industries may have a negative impact on the environment, while clean energy industries may contribute positively. The industrial structure is measured by the ratio of value-added of the tertiary industry to the value-added of the secondary industry (Wang and Luo, 2021).

Technological development: There is an interaction between technological development and ecological welfare. On the one hand, technological development may bring negative impacts such as environmental pollution and resource consumption. On the other hand, technological development can also provide innovative technologies and solutions to address environmental issues. Technological development is measured by the number of patents per GDP unit (Zeraibi et al., 2021).

Economic development: There exists a complex relationship between economic development and ecological welfare. On one hand, economic development may lead to negative impacts such as natural resource consumption and environmental pollution. On the other hand, a higher level of economic development may provide more resources and technologies to drive environmental protection actions. By controlling for economic development factors, the study can better explore the balance and coordination between government's economic development and ecological welfare. Economic development is represented by per capita GDP (Behjat and Tarazkar, 2021; Zhu and Zhang, 2014).

5. The conclusion and the abstract are two different sections, so it is recommended to summarize the core content of the conclusion section.

Based on the reviewers' suggestions, we revised the conclusion section, which reads as follows:

In the context of sustainable development, the central government has been continuously adjusting the assessment system for officials in order to achieve high-quality economic development. This study uses panel data samples and employs the super-efficiency SBM (Undesirable Outputs) to calculate the ecological welfare performance, exploring the impact of the assessment system on government competition and ecological welfare performance. Our research findings are as follows: ① Different government assessment systems have different effects on government competition and ecological welfare performance. Under the economic assessment system, government competition may have a negative impact on current and future ecological welfare performance, while the environmental assessment system can promote the improvement of ecological welfare performance. ② The spillover effects of government competition have significant differences in the impact on the ecological welfare performance of neighboring governments. Environmental competition in neighboring regions enhances local ecological welfare performance, while economic competition has a negative impact on local ecological welfare performance. ③ A multidimensional performance assessment system helps to alleviate the contradictions between government's economic development and environmental protection, short-term and long-term interests, and local and overall interests, and promotes the improvement of ecological welfare performance. After dynamically setting the weight of the assessment, government competition can promote the current and future improvement of ecological welfare performance. When the weight of economic and environmental assessments is in the range of 1:9 to 7:3, government competition contributes to the improvement of current ecological welfare performance. When the weight is in the range of 1:9 to 3:7, government competition contributes to the improvement of future ecological welfare performance. ④ Under the multidimensional performance assessment, the spillover effects of government competition are not significant, and the competition among neighboring governments does not affect local ecological welfare performance. At the same time, multidimensional competition can promote the improvement of local ecological welfare performance, solving the problems of economic competition inhibiting the growth of neighboring ecological welfare performance and its negative impact on local ecological welfare performance.

6. The policy recommendations are linked to the conclusions. However, the current policy recommendations seem to emphasize more on how to promote multidimensional performance appraisal, which is clearly not closely linked to the conclusions.

 Based on the reviewers' suggestions, we have revised the content of the policy recommendations, which are revised as follows:

First, introduce multidimensional performance assessment and integrate environmental assessment into the assessment system. When evaluating government performance, the balance between economy and ecology should be fully considered, focusing on long-term development and shared social value. Establish a scientific, fair, and actionable system of assessment indicators to ensure the scientific and fair assessment of government performance. Establish a clear and coordinated assessment mechanism to ensure the coordinated cooperation of various departments in fulfilling their environmental protection responsibilities and tasks.

Second, promote intergovernmental regional coordination and cooperation, establish regional cooperation mechanisms, strengthen multidimensional competition, and promote the improvement of local ecological welfare performance. The government needs to develop ecological environment protection plans, scientifically allocate environmental resources, increase investment in ecological environment protection, and achieve sustainable development. At the same time, efforts should be made to reduce the spillover effects of government competition to prevent the negative impact of economic competition among neighboring regional governments on local ecological welfare performance.

Third, in policy implementation, strengthen regulatory efforts, and pay attention to the construction of supporting assessment and incentive measures to ensure the effective implementation of policies. The government and enterprises should work together to establish a good cooperation model for environmental protection and sustainable development, promote the sustainable development of the economy and society, including public education, information disclosure, technological innovation, and environmental risk management. At the same time, the government needs to implement long-term, coordinated, and integrated policies to ensure the long-term and stable improvement of ecological welfare performance.

---

## [Decision Letter · Decision Letter 1]

27 Jul 2023

Multidimensional Performance Assessment, Government Competition and Ecological Welfare Performance

PONE-D-23-07357R1

Dear Dr. Yanqin,

We’re pleased to inform you that your manuscript has been judged scientifically suitable for publication and will be formally accepted for publication once it meets all outstanding technical requirements.

Kind regards,

Umer Shahzad, PhD

Academic Editor

PLOS ONE

Additional Editor Comments (optional):

Reviewers' comments:

Reviewer's Responses to Questions

**Comments to the Author**

1. If the authors have adequately addressed your comments raised in a previous round of review and you feel that this manuscript is now acceptable for publication, you may indicate that here to bypass the “Comments to the Author” section, enter your conflict of interest statement in the “Confidential to Editor” section, and submit your "Accept" recommendation.

Reviewer #1: All comments have been addressed

2. Is the manuscript technically sound, and do the data support the conclusions?

Reviewer #1: Yes

3. Has the statistical analysis been performed appropriately and rigorously? 

Reviewer #1: Yes

4. Have the authors made all data underlying the findings in their manuscript fully available?

Reviewer #1: Yes

5. Is the manuscript presented in an intelligible fashion and written in standard English?

Reviewer #1: Yes

6. Review Comments to the Author

Reviewer #1: (No Response)

7. PLOS authors have the option to publish the peer review history of their article (what does this mean?). If published, this will include your full peer review and any attached files.

Reviewer #1: No

---

## [Editor Report · Acceptance letter]

31 Jul 2023

PONE-D-23-07357R1 

Multidimensional Performance Assessment, Government Competition and Ecological Welfare Performance 

Dear Dr. Lv:

I'm pleased to inform you that your manuscript has been deemed suitable for publication in PLOS ONE. Congratulations! Your manuscript is now with our production department. 

Kind regards, 

on behalf of

Dr. Umer Shahzad 

Academic Editor

PLOS ONE